# The Effect of Governance on Industrial Wastewater Pollution in China

**DOI:** 10.3390/ijerph19159316

**Published:** 2022-07-29

**Authors:** Lili Li, Yaobo Shi, Yun Huang, Anlu Xing, Hao Xue

**Affiliations:** 1College of Humanities & Social Development, Northwest A&F University, Yangling, Xianyang 712100, China; 2School of Economics and Management, Xi’an University of Technology, Xi’an 710048, China; shiyaoboo@163.com (Y.S.); yunhuang@163.com (Y.H.); 3Meta Platforms, Inc., One Hacker Way, Menlo Park, CA 94025, USA; xinganlu1@gmail.com; 4Stanford Center on China’s Economy and Institutions, Stanford University, Stanford, CA 94305, USA; xuehjjx@gmail.com

**Keywords:** industrial wastewater governance, industrial wastewater pollution, fixed effect model, SYS-GMM

## Abstract

Water pollution not only aggravates the deterioration of the ecological environment and endanger human health, but also has a significantly negative impact on economic growth and social development. It is crucial to investigate the relationship between industrial wastewater governance and industrial wastewater pollution on the path to reduce water pollution. In this paper, we studied whether industrial wastewater governance affected industrial wastewater pollution using the panel fixed effect model and system generalized moment estimation model (SYS-GMM) with the panel data of 30 provinces from 2005 to 2020 in China. This is the only empirical analysis of the relationship between industrial wastewater governance and industrial wastewater pollution. We proxied industrial wastewater pollution by organic pollutants and inorganic pollutants and measured the per capita investment in industrial wastewater governance. The results shed light on the positive correlation between the per capita investment in industrial wastewater governance and industrial wastewater pollution. The increase in per capita investment in industrial wastewater governance promoted the increase of pollutant emissions from industrial wastewater. The estimation also indicated that there was an inverted U-shaped relationship between per capita GDP and inorganic /organic pollutants in industrial wastewater. Our empirical research shows that it is necessary to increase investment in industrial wastewater treatment and optimize the investment structure of environmental treatment, so as to pave the way for the comprehensive utilization of a variety of environmental treatment solutions.

## 1. Introduction

China’s economic development has made unprecedented achievements in recent years. However, the economic growth is still in the stage of extensive growth mode with high input and high environmental pollution. Water pollution is one of the serious environmental problems that China is currently facing [1]. If we keep the current economic growth model, on one hand, the situation of water environmental pollution will continue to deteriorate [2], on the other hand, water environmental pollution is affecting economic growth, and industrial wastewater emissions restrains economic growth [3]. Water pollution not only aggravates the deterioration of the ecological environment and endanger human health, but also has a great negative impact on social and economic development [4]. Industrial wastewater emission is an important part of wastewater emission and the amount of industrial wastewater emission increased rapidly in recent years. Especially, the amount of industrial wastewater emissions increased from 24.311 billion tons in 2005 to 735.32 billion tons in 2015 [5]. Therefore, the governance attaches great importance to industrial wastewater pollution. To improve the environmental quality of industrial wastewater, China has intensified the governance of industrial wastewater pollution, and the current work on environmental rule of law and environmental protection has made progress [6]. General Secretary Jinping Xi pointed out in his nineteenth important report that we should speed up the construction of ecological civilization and promote green development [7]. Under this background, it is very important to study the relationship between industrial wastewater governance and industrial wastewater pollution.

Environmental pollution has endangered the economic growth and environmental governance is one of the potential keys to mitigate the environmental pollution. It is, therefore, crucial to study the relationship between environmental governance and environmental pollution. The relationship between environmental governance and environmental pollution has attracted the attention of many scholars. Scholars’ conclusions on the effect of environmental governance are inconsistent. Some scholars believe that investment in environmental governance has a significant negative effect on the emission reduction of environmental pollutants. For example, some studies concluded that the total emission intensity of investment in environmental governance had a significant negative impact [8,9], that is, the greater the investment in governance, the stronger the emission. However, other scholars believe that investment in environmental governance has a positive effect on environmental pollution [10,11].For example, by using provincial panel data from 2003 to 2012, Gao (2015) concluded that environmental investment had a positive role in promoting industrial emission reduction in central and western China [12]. Some scholars also believe that the positive effect of environmental governance investment on environmental pollution is affected by factors such as governance structure and technology level [13,14]. 

There are many studies on the relationship between environmental governance and environmental pollution, including wastewater, waste gas, and solid pollutants [15,16,17]. However, previous studies neither analyzed the impact of environmental governance on industrial wastewater pollutants, nor further investigated the relationship between industrial wastewater governance and industrial wastewater pollutants. Till now, a few literatures investigated the environmental governance on water pollution by using the emission of industrial wastewater, chemical oxygen demand, ammonia nitrogen, and other major pollutants to measure water pollution [18,19]. However, few studies have been conducted on the relationship between investment in industrial wastewater governance and pollutant emission from industrial wastewater. In this paper, according to the chemical properties of industrial wastewater pollutants, we divide the emission of industrial wastewater pollutants into two categories, inorganic pollutants, and organic pollutants, to measure the industrial wastewater pollution. 

We hypothesize that investment in industrial wastewater treatment has a positive effect on its wastewater pollution. We have identified two channels through which investment in industrial wastewater treatment affects industrial wastewater pollution. The first is direct treatment. Through the use of investment funds in industrial wastewater treatment, enterprises and organizations who produced industrial wastewater purchase wastewater treatment equipment and improve the wastewater treatment equipment system. They also improve the recovery rate of water purification, and advanced treatment of sewage through the development of new wastewater treatment technology, such as research on more efficient, more economic, and energy-saving treatment technology. The second effect channel is through indirect treatment. Both economic growth and environmental investment have positive effects on each other’s production. That is to say, the increase in investment in industrial wastewater treatment has promoted the growth of the national economy, and the growth of the national economy has led to the pursuit of higher quality environment [20,21]. It invests in industrial wastewater treatment increase and promotes the intensity of wastewater treatment. 

This paper contributes to the existing literature from several aspects. Firstly, we provide a new perspective for the empirical study of environmental economics. As far as we know, this is the only empirical analysis of the relationship between industrial wastewater governance and industrial wastewater pollution. Secondly, we use static panel and dynamic panel estimation methods to verify the relationship between industrial wastewater governance and wastewater pollutants by employing data from 30 provinces from 2005 to 2020. We use the panel fixed effect model to verify the impact of industrial wastewater governance on industrial wastewater pollutants, however, the estimated results may be inconsistent. To solve this problem and consider the dynamic factors, we propose a system generalized moment method (SYS-GMM) to further study the relationship between industrial wastewater governance and industrial wastewater pollutants. Thirdly, to further verify the robustness of the relationship between industrial wastewater investment and industrial wastewater pollutants, we introduced the proportion of industrial wastewater governance investment among total environmental governance investment to measure industrial wastewater governance. For the rigor of estimation, this study also introduces several control variables, such as industrialization, per capita Gross Domestic Product (GDP), foreign direct investment (FDI), population density, and so on. Empirical results show that there is a positive correlation between industrial wastewater governance and wastewater pollutants. Increased investment in industrial wastewater governance will increase the content of inorganic and organic pollutants in industrial wastewater. The proportion of industrial wastewater governance investment in environmental governance investment is negatively correlated with pollutants in industrial wastewater. Our findings provided empirical research results and recommendations for policymakers to improve wastewater pollution in transition countries. 

The following structure of the paper is as follows: Section 2 is methodology and data. Section 3 is the main empirical results. The Section 4 is the discussion focused on strategies. The last section is conclusion of the article and relevant policy recommendations.

## 2. Materials and Methods

### 2.1. Data Sources

Considering the continuity and availability of data, this study used the panel data of 30 provinces in China from 2005 to 2020. Other provinces were not included because their data are incomplete(study samples are excluding Hong Kong, Macao and Taiwan, and excluding Tibet with serious data loss).The data mainly came from China Statistical Yearbook, China Environmental Yearbook, and China Industrial Economic Statistics Yearbook. Each record represents a data point for a province in a year.

### 2.2. Outcome Measures

Industrial wastewater pollution was measured by the emissions of industrial wastewater. According to the statistics of industrial wastewater pollutants in the China Environmental Yearbook, industrial wastewater pollutants can be divided into ten kinds of pollutants, including mercury, cadmium, hexavalent chromium, lead, arsenic, volatile phenol, chloride, chemical oxygen demand, and the total amount of petroleum and ammonia nitrogen emissions. This paper divided these ten kinds of pollutants into two categories according to their chemical properties. The first category was defined as inorganic pollutants. It includes Mercury, cadmium, hexavalent chromium, lead, and arsenic in industrial wastewater pollutants. We summed up these five pollutants as the value of inorganic pollutants. The second category was defined as organic pollutants. We took the total amount of volatile phenol, chloride, chemical oxygen demand, petroleum, and ammonia nitrogen in industrial wastewater as the value of organic pollutants. In this paper, we used inorganic pollutants and organic pollutants to measure industrial wastewater pollution.

Industrial wastewater governance was measured by the per capita investment of industrial wastewater governance. It is obtained from the industrial wastewater investment of each province divided by the population of each province at the end of the year. To check the robustness of the empirical results, we further adopted the proportion of industrial wastewater governance investment as an explanatory variable. It is obtained from investment in industrial wastewater governance divided by total investment in environmental governance.

### 2.3. Covariates

To estimate the relation more robustly, this study also included a couple of control variables including GDP, industrial structure, foreign direct investment (FDI), trade openness, population density, and urbanization level. GDP is usually the main indicator of a country’s overall economic growth. According to previous studies [22,23], this paper used per capita GDP to measure the economic growth of each province. Provinces in China have heterogenous levels of economic development, resulting in various industrial structures and wastewater pollutants. Since the change in industrial structure affects the content of pollutants in wastewater to a certain extent, we measure industrial structure by dividing the output value of the secondary industry by GDP. Foreign investment is considered as an indicator of advanced production technology. On the other hand, foreign investment with poor environmental performance flows to areas with weak environmental control. This paper took the proportion of foreign direct investment in GDP as the degree of foreign investment in each province [24,25]. Trade openness is the basic index to measure the export-oriented degree of provincial economic development, which is conducive to the economic growth and transformation of various provinces [26,27]. However, the improvement of trade openness will lead to the expansion of the economic scale and an increase in pollution emissions. According to Jayanthakumaran and Liu (2012) [28], we used the proportion of total imports to the GDP of each province to express trade openness. The original unit of trade data is the US dollar. In this paper, we converted it into CNY (the abbreviation of international standard Chinese yuan) according to the annual average exchange rate. The population density was the main source of environmental pollution to a large extent [29,30]. According to previous studies, the population density of each province is taken as one of the control variables, and logarithmic processing is done. The acceleration of urbanization promotes the expansion of urban scale, thus affecting environmental quality [31,32]. According to the existing research, this paper used the proportion of the urban population at the end of that year to measure the urbanization rate. Table 1 gives a detailed description of variables and data sources.

### 2.4. Statistical Analysis

The overarching goal of our research is to investigate the impact of industrial wastewater governance on industrial wastewater pollution by using panel data from 2005 to 2020. Considering the different scale in numerical values among different variables, we took the natural logarithm of inorganic pollutant emissions, organic pollutant emissions, per capita industrial wastewater investment, per capita GDP, and population density. Therefore, this paper established the following panel data model:(1)$$\log(P_{i,t})=\alpha_{0}+\alpha_{1}\log\left({\mathrm{Investment}}_{i,t}\right)+{\gamma Z}_{i,t}+\delta_{i}+\eta_{i}+\varepsilon_{i,t}$$

In Equation (1), $P_{i,t}$ denotes inorganic or organic pollutants for measuring the industrial wastewater pollution; ${\mathrm{Investment}}_{i,t}$ denotes the per capita investment in industrial wastewater governance; $Z_{i,t}$ is a vector of control variables that may affect industrial wastewater pollution. i denotes provinces, i = 1, 2… 30; t means time, t = 2005…, 2020, $\delta_{i}$ and $\eta_{i}$ denote the fixed effect variables of time and region, respectively; and $\varepsilon_{i,t}$ is the error term.

However, the pollution of industrial wastewater is a dynamic adjustment process. The results of the previous period have a certain impact on the current results. However, the static panel data model can’t reflect the lag effect of the dependent variables. To solve this problem, Arellano and Bond (1991) proposed the differential GMM (Diff-GMM) estimator [33]. The basic idea is to use the lag explanatory variable in the differential equation as the tool variable to estimate. However, the Diff-GMM estimator may occur the problem of weak exogenous instrument variables. To solve the problem of the weak instrument, Arellano and Bover (1995) and Blundell and Bond (1998) put forward the system GMM (SYS-GMM) estimator [34,35]. The system GMM uses both horizontal and difference equations to construct tool variables, which increases the capacity of sample tool variables and can effectively solve the problem of weak tool variables. Based on the different selection of weight matrix, GMM estimation can be divided into one-step estimation and two-step estimation. Bond et al. (2001) argued that two-step GMM estimation could better deal with autocorrelation and heteroscedasticity problems with limited samples [36]. Therefore, we used a two-step GMM estimation method to better analyze the impact of industrial wastewater governance investment on industrial wastewater pollution. The SYS-GMM estimation model of the system is as follows:(2)$$\log(P_{i,t})=\alpha_{0}+\alpha_{1}\log\left(P_{i,t-1}\right)+\alpha_{2}\log\left({\mathrm{Investment}}_{i,t}\right)+{\gamma Z}_{i,t}+\delta_{i}+\eta_{i}+\varepsilon_{i,t}$$

In Equation (2), $P_{i,t-1}$ denotes the lag term of the dependent variable, while the rest of the variables have the same meaning as Equation (1).

In addition, for the validity of GMM estimation results, Arellano and Bover (1995) and Blundell and Bond (1998) put forward a test method, that is, there is no second-order sequence correlation for the random error term after one difference, but there can be a first-order autocorrelation. Arellano-Bond is usually used to test, and its original assumption is that there is no second-order random error sequence autocorrelation [34,35]. At the same time, we used the Sargan test whether the instrumental variables used in the model exist over-recognition [37]. The original assumption of the Sargan test is that there is no over-recognition of the instrumental variables.

To check the robustness of industrial wastewater treatment on industrial wastewater pollution, we further use the ratio of industrial wastewater treatment investment to environmental treatment investment, namely the variable Rate, to measure industrial wastewater treatment. The panel fixed effect model is as follows:(3)$$\log(P_{i,t})=\alpha_{0}+\alpha_{1}Rate_{i,t}+{\gamma Z}_{i,t}+\delta_{i}+\eta_{i}+\varepsilon_{i,t}$$

In Equation (3), $P_{i,t}$ denotes inorganic or organic pollutants for measuring the industrial wastewater pollution; $Rate_{i,t}$ denotes investment in industrial wastewater governance divided by total investment in environmental governance; $Z_{i,t}$ is a vector of control variables that may affect industrial wastewater pollution. i denotes provinces, i = 1, 2… 30; t means time, t = 2005…, 2020, $\delta_{i}$ and $\eta_{i}$ denote the fixed effect variables of time and region, respectively; and $\varepsilon_{i,t}$ is the error term.

The panel SYS-GMM model is Model 4 as follows:(4)$$\log(P_{i,t})=\alpha_{0}+\alpha_{1}\log\left(P_{i,t-1}\right)+\alpha_{2}Rate_{i,t}+{\gamma Z}_{i,t}+\delta_{i}+\eta_{i}+\varepsilon_{i,t}$$

In Equation (4), $P_{i,t-1}$ denotes the lag term of the dependent variable, while the rest of the variables have the same meaning as Equation (3).

## 3. Results

### 3.1. Descriptive Statistics

Table 2 is a statistical description of each variable. As we can see, the average of the Investment (per capita investment of industrial wastewater governance) is 230 yuan(CNY). However, there is a big gap in industrial wastewater governance across provinces, according to the difference between the maximum investment value and the minimum investment value. The maximum and minimum of Rate are 0.64 and 0, and the average value is 0.02, indicating that the proportion of industrial wastewater governance investment to environmental governance investment in each province is quite different. The mean of Industry is 0.37, and the maximum value is 0.53, which indicates that the proportion of industrial added value in GDP in China’s provinces is relatively high. The average value of GDP is 2.0 × 10^4^, and the maximum value is 1.1 × 10^5^, which implies that there is still much room for the per capita income of each province to rise. From the perspective of FDI, the average value of FDI in China is 0.003 and the maximum value is 0.01, which shows that the proportion of foreign direct investment in GDP varies greatly across regions. The gap between the maximum value and the minimum value of Openness is large, which indicates that there are great differences in the degree of economic openness among provinces in China. As for Density, there is a big gap between the maximum value and the minimum value, which indicates that the population of different provinces in China is different. In addition, from the level of Urbanization, the average urbanization rate in China is 0.55, and the maximum value is 0.9, indicating that there is a large space for the improvement of the urbanization rate in China.

Figure 1 and Figure 2 (in the end of the document) show the emissions of inorganic and organic pollutants in 30 provinces from 2005 to 2020 respectively. From Figure 1, we can see that the time trend of inorganic pollutant emissions in different provinces is different and fluctuates greatly. Some provinces show a steady downward trend, such as Hunan and Jiangsu, while others show an upward trend, such as Jilin and Inner Mongolia. From Figure 2, we can see that most of the time trends of organic pollutant emissions in different provinces show a downward trend, with little fluctuation. Some provinces are stable, such as Hebei and Hunan, while some other provinces fluctuate greatly, such as Xinjiang, Gansu and Qinghai. Some provinces are on the rise, such as Tianjin, Jiangsu etc.

### 3.2. Estimated Results: Panel Fixed Effect Model

F-test and Houseman test was applied to the interpreted variables Inorganic and Organic in the static panel model respectively. In the Hausman test, all models in Table 3 rejected the original hypothesis of random effect at the level of 10%. Therefore, this paper adopted the panel fixed effect model to analyze the relationship between industrial wastewater governance and industrial wastewater pollution.

Firstly, we present the results of the panel model in Table 3. As shown in Table 3, the dependent variables used in this study are the emissions of inorganic pollutants and organic pollutants to measure the pollution of industrial wastewater. In the process of regression analysis, we introduced control variables into the model respectively. Model 1 showed the effects of the per capita governance investment of industrial wastewater, industrialization rate, per capita GDP, and square per capita GDP on inorganic pollutant emissions. Model 2 added FDI, and Openness based on model 1. Model 3 added population density and urbanization rate based on model 2. Empirical results showed that for inorganic pollutants regression, no matter what control variables are added, the coefficient of the investment in industrial wastewater governance was positive and statistically significant at 1% level. This indicated that the investment in industrial wastewater governance affect the reduction of industrial wastewater pollution. Model 4 showed the effect of the per capita governance investment of industrial wastewater, industrialization rate, per capita GDP, and square per capita GDP on organic pollutant emissions. Model 5 added FDI and Openness based on model 4. Model 6 added population density and urbanization rate based on model 3. The estimation results indicated that the coefficient of the investment in industrial wastewater governance was positive and significant at 1% level regardless of adding any control variables. That’s proving that when the per capita investment in environmental governance increases, the emissions of both inorganic and organic pollutants from industrial wastewater also increase. 

As for control variables, through the empirical results of model 1–3, we found the coefficient of the logarithm of GDP per capita was significantly negative at the level of 1%, indicating that economic growth was conducive to significantly reducing the emissions of inorganic pollutants in wastewater and improving the quality of wastewater, and the coefficient of the square of the logarithm of per capita GDP is significantly positive at the level of 5–10%, which indicates that the logarithm of per capita GDP has a U-shaped relationship with inorganic pollutants in wastewater, it coincided with the economic phenomena described by the Environmental Kuznets Curve. The coefficient of Density was significantly positive at the level of 10% only in model 3, indicating that the increase of Density would increase the emissions of inorganic wastewater.

Through model 4–6, we found that the coefficient of the industry variable was significant and negative at the level of 1%, indicating that with the increase of the proportion of industrial added value to GDP, the emissions of organic pollutants in wastewater decreased significantly. The coefficient of the logarithm of per capita GDP was significantly positive at the 1% level, implying that economic growth was conducive to a significant increase in organic pollutant emissions from wastewater, while the square coefficient of per capita GDP was significantly negative at the 1% level, indicating that the logarithm of per capita GDP had an inverted U-shaped relationship with organic pollutant emissions. The coefficient of openness was significantly negative at 1–5% level, which indicated that the increase of openness significantly reduced the emissions of organic pollutants from wastewater. The coefficient of FDI, openness and density was significantly negative at 1–5% level, which indicated that the increase of FDI, openness and density significantly reduced the emissions of organic pollutants from wastewater. The coefficient of Urbanization was significantly positive at the level of 1% only in model 3, indicating that the increase of Urbanization would increase the emissions of organic wastewater. In addition, empirical estimate show that other variables have no significant relationship with inorganic and organic pollutants in Table 3.

### 3.3. Estimated Results: SYS-GMM Model

From the dynamic panel data estimation results of model 1–6 in Table 4, it can be found that the Arellano-Bond second-order random error sequence autocorrelation test (AR(2)) results of model 1–6 accept the original hypothesis that the perturbation term is not autocorrelated, the Sargan test results accepted the original hypothesis that “all the instrumental variables are valid”, which indicates that the GMM estimation results of the system are valid.

According to the results of Table 4, we can draw the following conclusions: Firstly, the coefficients of lagged of the inorganic pollutants and organic pollutants are positive and statistically significant in the regression results of all models. It shows that the emission of wastewater pollutants is persistent and dynamic. Secondly, empirical results showed that for inorganic or organic pollutants regression, no matter what control variables are added, the coefficient of the investment in industrial wastewater governance was positive but not statistically significant. That’s proving that the positive correlation between the per capita investment in industrial wastewater governance and industrial wastewater both inorganic and organic pollutants. Thirdly, the results of the industrialization rate in Table 4 are consistent with Table 3, which shows the robustness of empirical results. Fourth, the relationship between FDI and industrial wastewater pollutants is negative, which means that the increase in FDI will reduce organic pollutants and inorganic pollutants in wastewater.

### 3.4. Robustness Check

To study the robustness of industrial wastewater treatment on industrial wastewater pollution, we further use the ratio of industrial wastewater treatment investment to environmental treatment investment, namely the variable Rate, to measure industrial wastewater treatment. The panel fixed effect regression is shown in Table 5 and the panel SYS-GMM regression is shown in Table 6.

The empirical results in Table 5 are as follows. Firstly, the variable of Rate in model 4–6 is significantly negative at 1% level, which indicates that the lower the proportion of industrial wastewater governance investment in environmental investment, the more serious the organic pollution in industrial wastewater. While the variable of Rate in model 1–3 is not significant. Secondly, the results of Table 5 verify that the logarithm of GDP per capita has a U-shaped relationship with the emission of inorganic pollutants in wastewater and an inverted U-shaped relationship with the discharge of organic pollutants in wastewater. Thirdly, the results of other control variables are consistent with the regression results in Table 3.

The estimated results of SYS-GMM are shown in Table 6. From the estimation results in Table 6, it can be found that the Arellano-Bond second-order random error sequence autocorrelation test (AR (2)) results of model 1–6 accept the original assumption that the perturbation term has no autocorrelation, and the Sargan test results accept the original assumption that all the instrumental variables are valid, which indicates that the GMM estimation results of the system are valid.

According to the results of Table 6, we can draw the following conclusions: Firstly, the coefficients of the lagged value of inorganic pollutants and organic pollutants are significantly positive at the level of 1%, indicating that the emissions of wastewater pollutants have significant sustainability and dynamics. In other words, wastewater pollution and governance are long-term processes. Secondly, the variable of Rate in model 2 and model 4–6 is significantly negative at 1–5% level, which indicates that the lower the proportion of industrial wastewater governance investment in environmental investment, the more serious the inorganic or organic pollution in industrial wastewater. While the variable of Rate in model 1 and 3 is not significant but negative, although it is not statistically significant, it also shows that the increase in the proportion of industrial wastewater governance investment in environmental governance investment can promote the reduction of wastewater pollutant emissions in the long run. Thirdly, the estimated results of the remaining variables are consistent with those in Table 4, which confirms the robustness of the empirical results.

## 4. Discussion

To analyze the impact of industrial wastewater governance on industrial wastewater pollution in China, this paper used provincial panel data from 2005 to 2020, took per capita investment in industrial wastewater governance as an indicator of industrial wastewater governance, and used the panel fixed effect model and system GMM regression to verify the relationship between investment in industrial wastewater governance and wastewater pollutant emissions. The results show that in the short run, there is a significant positive relationship between the per capita investment in industrial wastewater treatment and the organic and inorganic pollutants in wastewater. That is, the per capita investment in industrial wastewater increased by 1%, the content of inorganic pollutants increased by 0.034%, and the content of organic pollutants increased by 0.10%. But in the long run, the effect of wastewater treatment is different, that is, the per capita investment in industrial wastewater treatment increased by 1%, the content of inorganic pollutants increased by 0.18%, and the content of organic pollutants increased by 0.05%. The empirical results show that the per capita investment in wastewater treatment has a positive effect on promoting organic or inorganic pollutants in wastewater in the short and long run. 

To sum up, industrial wastewater governance has not contributed to reducing industrial wastewater pollution but also has increased industrial wastewater pollution to some extent. The view that investment in environmental governance has a positive impact on wastewater pollution has also been put forward by some scholars, for example, Gao et al. (2015) constructed a pollution intensity index of industrial wastewater and industrial waste gas [12], and found that although the response of pollution reduction intensity to environmental protection investment in different regions is different, overall, environmental protection investment has a positive impact on industrial pollution reduction. The possible reasons are: on the one hand, the total investment in environmental governance is insufficient [38,39]; on the other hand, the investment structure of environmental governance is unreasonable [40], the proportion of investment in urban environmental infrastructure construction is relatively high, and the investment in industrial pollution source control and construction project “three simultaneous” environmental protection needs to be further strengthened [41].Therefore, the government should invest more in the treatment of organic pollutants in industrial wastewater. 

## 5. Conclusions

Investment in industrial wastewater is the basis of industrial wastewater governance in China. There is little literature research on the impact of investment in industrial wastewater governance on industrial wastewater pollution. This is the only empirical analysis of the relationship between industrial wastewater governance and industrial wastewater pollution. The results shed light on the positive correlation between the per capita investment in industrial wastewater governance and industrial wastewater pollution. The increase in per capita investment in industrial wastewater governance promoted the increase of pollutant emissions from industrial wastewater.

The results also indicate that the logarithm of per capita GDP has a U-shaped relationship with inorganic pollutants in wastewater, it coincided with the economic phenomena described by the Environmental Kuznets Curve. That reflects there are insufficient investments in environmental governance in most parts of China currently. According to the experience of developed countries, investment in environmental protection should account for 3% of GDP [42,43]. However, investment in environmental governance in 30 provinces in China was 1.34% on average from 2005 to 2020, and the proportion of wastewater governance in total environmental governance is very small. 

On the whole, increasing the proportion of industrial wastewater treatment investment has obvious effect on water pollution treatment, while China has always had the problem of small proportion of industrial pollution source treatment investment. From 2005 to 2020, the average proportion of investment in industrial wastewater governance was only 2 percent. Therefore, how to optimize the investment structure of environmental treatment, strengthen the treatment of organic pollutants, and strengthen the treatment and monitoring of pollution sources should be an important direction of wastewater quality treatment.

From the empirical results, to effectively treat industrial wastewater and to reduce the content of pollutants in industrial wastewater, the government should increase the proportion of wastewater governance investment. In addition to environmental investment, the government has many other ways to reduce wastewater pollution, such as implementing relevant water environmental protection policies and encourage the development of green environmental protection industry. The government can also use a variety of techniques to reduce wastewater emissions, including enacting laws and regulations, collecting environmental taxes and fees, promoting emissions trading, and so on.

## Figures and Tables

**Figure 1 ijerph-19-09316-f001:**
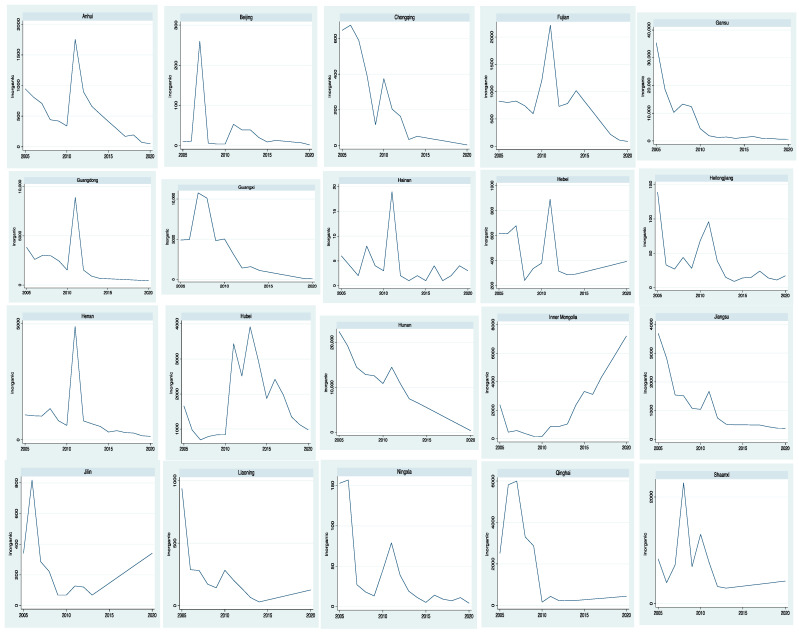
The emissions of inorganic pollutants.

**Figure 2 ijerph-19-09316-f002:**
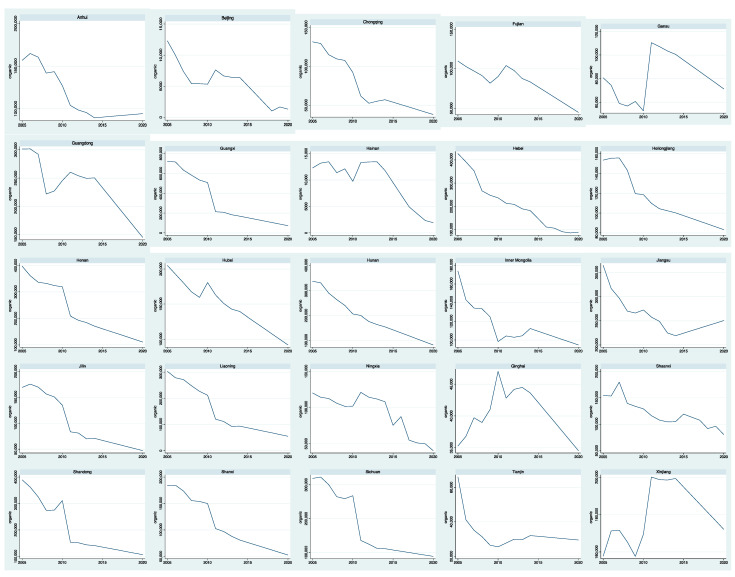
The emissions of organic pollutants.

**Table 1 ijerph-19-09316-t001:** Definition of variables and data sources.

Variables	Definition	Source
Inorganic	Emissions of inorganic pollutants	1
Organic	Emissions of organic Pollutants	1
Investment	Per capita investment in industrial wastewater governance	1
Rate	Investment in industrial wastewater governance divided by total investment in environmental governance	1
Industry	Industrial added value divided by GDP	2
GDP	GDP divided by the population at the end of the year	2
FDI	Foreign direct investment (FDI) divided by GDP	2
Openness	Total export-import volume divided by GDP	2
Density	The resident population at the end of the year divided by provincial area	3
Urbanization	An urban population divided by total population	2

Note: Reference numbers in source are as follows: 1 = China Environmental Yearbook (2005–2020); 2 = China Statistical Yearbook (2005–2020); 3 = Statistical Yearbook of China’s Industrial Economy (2005–2020).

**Table 2 ijerph-19-09316-t002:** Descriptive statistics of variables in the model.

Variable	Observations	Mean (SD)	Min	Max
Inorganic	480	1.3 (3.1) × 10^3^	1	3.5 × 10^4^
Organic	480	1.3 (1.0) × 10^5^	1028	7.2 × 10^5^
Investment	480	230 (201)	5.3	1416
Rate	480	0.02 (0.05)	0	0.64
Industry	480	0.37 (0.08)	0.1	0.53
GDP	480	2.0 (1.8) × 10^4^	543	1.1 × 10^5^
FDI	480	0.003 (0.002)	0	0.01
Openness	480	0.30 (0.37)	0.01	1.78
Density	480	412 (512)	7.59	3061
Urbanization	480	0.55 (0.14)	0.27	0.90

Data source: China Environmental Yearbook (2005–2020); China Statistical Yearbook (2005–2020); Statistical Yearbook of China’s Industrial Economy (2005–2020).

**Table 3 ijerph-19-09316-t003:** Estimation Results of fixed effect model.

Variables	Log (Inorganic)			Log (Organic)		
	Model 1	Model 2	Model 3	Model 4	Model 5	Model 6
Log(Investment)	0.32 ***	0.33 ***	0.34 ***	0.10 **	0.13 ***	0.10 **
	(0.10)	(0.11)	(0.11)	(0.04)	(0.04)	(0.04)
Industry	1.46	1.49	1.74	−1.95 ***	−1.85 ***	−1.86 ***
	(1.29)	(1.29)	(1.30)	(0.52)	(0.52)	(0.51)
Log(GDP)	−3.59 ***	−3.66 ***	−4.02 ***	1.65 ***	1.70 ***	1.89 ***
	(0.75)	(0.78)	(0.81)	(0.30)	(0.31)	(0.31)
(Log(GDP))^2^	0.07 **	0.08 *	0.09 **	−0.06 ***	−0.06 ***	−0.88 ***
	(0.06)	(0.04)	(0.04)	(0.01)	(0.02)	(0.02)
FDI		−10.3	−5.82		−24.1 **	−26.6 **
		(26.9)	(26.9)		(10.74)	(10.48)
Openness		0.07	0.62		−0.14	−0.85 ***
		(0.33)	(0.49)		(0.13)	(0.19)
Log(Density)			1.96 *			−1.04 **
			(1.04)			(0.41)
Urbanization			0.56			3.52 ***
			(2.43)			(0.95)
Constant	30.1 ***	30.34 ***	21.44 ***	2.53	2.26	6.63 ***
	(4.03)	(4.15)	(6.31)	(1.62)	(1.66)	(2.46)
Observations	480	480	480	480	480	480
R^2^	0.39	0.39	0.40	0.51	0.52	0.55
Prob > F	0.000	0.000	0.000	0.000	0.000	0.000

Notes: 1. The values in parentheses denote the standard errors. * indicates significance at 10%. ** indicates significance at 5%. *** indicates significance at 1%. 2. Prob > F refers to the *p* value. The probability is less than 0.05, which means significant.

**Table 4 ijerph-19-09316-t004:** Estimation Results of GMM model.

Variables	Log (Inorganic)			Log (Organic)		
	Model 1	Model 2	Model 3	Model 4	Model 5	Model 6
L.log (P)	0.85 ***	0.83 ***	0.69 ***	0.94 ***	0.96 ***	0.87 ***
	(0.09)	(0.09)	(0.11)	(0.03)	(0.05)	(0.07)
Log (Investment)	0.07	0.09	0.18	0.03	0.03	0.05
	(0.33)	(0.34)	(0.49)	(0.04)	(0.04)	(0.04)
Industry	1.70	2.44	0.39	0.92 ***	0.87 *	0.60
	(2.28)	(2.78)	(1.24)	(0.34)	(0.50)	(0.58)
Log (GDP)	−0.95	−0.83	0.63	−0.74	−0.64	−0.19
	(2.70)	(3.83)	(3.74)	(0.69)	(0.57)	(0.43)
Log (GDP)^2^	0.04	0.04	−0.04	0.03	0.03	0.01
	(0.15)	(0.21)	(0.21)	(0.03)	(0.03)	(0.02)
FDI		−0.02	−0.10		−0.04	−0.02
		(0.20)	(0.40)		(0.04)	(0.04)
Openness		−0.22	−0.13		0.09	0.09
		(0.16)	(0.30)		(0.04)	(0.06)
Log (Density)			0.18			−0.06
			(0.61)			(0.06)
Urbanization			−2.07			−0.88 ***
			(1.85)			(0.29)
Constant	6.01 ***	3.63	−4.29	3.85	3.13	1.71
	(2.29)	(17.2)	(21.51)	(3.34)	(2.77)	(2.16)
AR(2)	0.182	0.198	0.214	0.115	0.129	0.102
Sargan	0.171	0.119	0.103	0.961	0.969	0.832

Notes: 1. The values in parentheses denote the standard errors. * Indicates significance at 10%. *** indicates significance at 1%. 2. AR(2) means Arellano-Bond second-order random error sequence autocorrelation test. 3. Sargan means Sargan Test. 4. L.log (P) denotes the lag term of the dependent variable, that is lagged of the inorganic pollutants and organic pollutants.

**Table 5 ijerph-19-09316-t005:** The results of panel fixed effect model: Rate as the independent variable.

Variables	Log (Inorganic)			Log (Organic)		
	Model 1	Model 2	Model 3	Model 4	Model 5	Model 6
Rate	0.87	1.06	0.96	−2.69 ***	−3.00 ***	−2.74 ***
	(0.88)	(0.92)	(0.94)	(0.33)	(0.33)	(0.34)
Industry	1.96	1.87	2.15	−1.57 ***	−1.41 ***	−1.46 ***
	(1.29)	(1.30)	(1.31)	(0.48)	(0.48)	(0.47)
Log(GDP)	−3.09 ***	−3.30 ***	−3.60 ***	2.09 ***	2.43 ***	2.47 ***
	(0.74)	(0.79)	(0.81)	(0.27)	(0.29)	(0.29)
Log(GDP)^2^	0.06	0.07 *	0.07 *	−0.08 ***	−0.10 ***	−0.11 ***
	(0.04)	(0.04)	(0.04)	(0.01)	(0.01)	(0.01)
FDI		4.41	8.41		−11.2	−15.1
		(26.55)	(26.71)		(9.77)	(9.64)
Openness		0.26	0.58		−0.42 ***	−0.96 ***
		(0.34)	(0.50)		(0.13)	(0.18)
Log(Density)			1.73			−0.73 *
			(1.05)			(0.38)
Urbanization			1.96			3.00 ***
			(2.44)			(0.88)
Constant	27.77 ***	28.70 ***	20.62 ***	0.33	−1.22	2.25
	(3.99)	(4.17)	(6.47)	(1.49)	(1.53)	(2.33)
Observations	480	480	480	480	480	480
R2	0.38	0.38	0.38	0.57	0.58	0.60
Prob > F	0.000	0.000	0.000	0.000	0.000	0.000

Notes: 1. The values in parentheses denote the standard errors. * indicates significance at 10%. *** indicates significance at 1%. 2. Prob > F refers to the *p* value. The probability is less than 0.05, which means significant.

**Table 6 ijerph-19-09316-t006:** The results of SYS-GMM model: Rate as independent variable.

Variables	Log (Inorganic)			Log (Organic)		
	Model 1	Model 2	Model 3	Model 4	Model 5	Model 6
L.log (P)	0.83 ***	0.70 **	0.67 ***	0.68 ***	0.70 ***	0.75 ***
	(0.08)	(0.10)	(0.11)	(0.13)	(0.11)	(0.08)
Rate	−0.09	−0.17 **	−0.02	−0.13 ***	−0.11 **	−0.06 **
	(0.06)	(0.08)	(0.08)	(0.05)	(0.05)	(0.03)
Industry	0.52 *	0.74	0.30	0.72 **	0.66 **	0.40 ***
	(0.31)	(0.53)	(0.60)	(0.30)	(0.26)	(0.15)
Log(GDP)	−0.59	−0.80	−0.91	0.95	0.35	0.38
	(2.43)	(2.88)	(2.37)	(0.62)	(0.40)	(0.45)
Log(GDP)^2^	0.03	0.04	0.06	−0.05	−0.02	−0.02
	(0.13)	(0.15)	(0.12)	(0.03)	(0.02)	(0.03)
FDI		0.10	0.06		−0.06	−0.03
		(0.16)	(0.12)		(0.04)	(0.03)
Openness		−0.06	0.15		−0.02	0.05
		(0.16)	(0.20)		(0.05)	(0.04)
Log(Density)			−0.16			−0.01
			(0.17)			(0.04)
Urbanization			−2.14			−0.81 ***
			(1.47)			(0.28)
Constant	3.94	4.21	5.10	−0.11	1.83	0.57
	(11.64)	(14.11)	(12.23)	(0.03)	(1.96)	(2.24)
AR(2)	0.183	0.197	0.191	0.167	0.231	0.103
Sargan	0.9402	0.9428	0.9561	0.9679	0.974	0.679

Notes: 1. The values in parentheses denote the standard errors. * Indicates significance at 10%. ** indicates significance at 5%. *** indicates significance at 1%. 2. AR(2) means Arellano-Bond second-order random error sequence autocorrelation test. 3. Sargan means Sargan Test resutls. 4. L.log (P) denotes the lag term of the dependent variable, that is lagged of the inorganic pollutants and organic pollutants.

## Data Availability

Data are available upon request.

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
