# Peer review of "The Effect of Governance on Industrial Wastewater Pollution in China"

_ijerph, 2022, doi:10.3390/ijerph19159316_

Round 1
Reviewer 1 Report
Overall, this is a well-written piece which considers an issue of importance. Despite the potential unique selling points of this piece and contribution to the field of environmental governance, the value of the piece is somewhat undersold. More grounding as to the significance of this issue is needed (i.e. what is the significant problem? How is this research key to highlighting some of these issues?). The background to the problem is not well-referenced (see my line by line comments) and more grounding is needed. The analysis of the data is quite limited and the findings are not well-explained (see my line by line suggestions). The research is also based on data which is rather old, so greater justification is needed for not using more recent documents for the research (i.e. is there a causal link between these documents and existing policy). It is also unclear what existing policy is and how this piece hopes to influence it. Although the data is interesting and it's an important piece of work, more grounding is needed in the introduction and analysis needs to be more detailed throughout, with robust findings and suggestions (either for future work or specific policy changes) within the conclusion.
Lines 25-26: Rephrase final sentence, e.g. ‘our empirical research pointed to the need for…’ ‘paved the way for…’ etc.
Line 34 – although ‘growth’ refers to economic growth, it is too general as to why this is a problem, especially to non-specialists in this area. It’s necessary to clearly define economic growth for the purposes of this piece (since this is a broad term) and identify the specific problem and causal connection between the two.
Line 42 – ‘environmental quality’ of what?
Line 45 – General Secretary reference needs to be cited here.
Line 54 – ‘some’ scholars implies more than one. Some referencing needed here.
Line 65 – refers to ‘many’ studies but none of these are referenced.
Line 78 – grammar, words missing here re effect of industrial wastewater.
Line 90 – refers to regional economy. Are there any references here? Need to distinguish between regional and national economy.
Line 94 – unique selling point here needs to be made prominent within the abstract.
Lines 121-124 – research seems to be based on quite old resources. Details here say that others are not used because ‘incomplete’. Incomplete from 2015 onwards? Because the research is largely based on quite old data, more justification for the value of the sources would be required.
Lines 141-144 – not clear. Further explanation required.
Line 149 – economic growth rather than level? Need for prevision, consistency in terminology.
Line 165 – RMB? Needs explanation.
General comments: Although graphical representation of data is effective and explained well, there needs to be more discussion about what we can learn from all of this.
Conclusions – quite short and vague. I would like to see a more detailed and succinct overview of the conclusions from the findings and while detailed explanation of strategies is not essential in a piece of this nature, some clearer ideas ought to be given.
Reviewer 2 Report
1. Suggestions to improve...
1.1 Missing acronym definitions:
Gross domestic product (GDP)
Renminbi (RMB)
Autoregression (AR)
--------
1.2 Improve Table 1
Source -> put respective reference as number
and enlarge column 2 to accommodate more text
--------
1.3 Replace Table 2 by histograms?
> better visual overall information
> avoid statistical problems... e.g. in Inorganic data Mean = 1634.26 and Std.Dev. = 3541.77
this corresponds to 1.6(3.5) x 10³ which means values may be negative?
> recommendations for parameter estimates representation...
uncertainty with a maximum of 2 significant digits and central estimate rounded to same decimal places
Mean = 1634.26 and Std.Dev. = 3541.77 -> should be represented as 1.6(3.5) x 10³
--------
1.4 Tables 3, 4 and 5
> recommendations for parameter estimates representation...
uncertainty with a maximum of 2 significant digits and central estimate rounded to same decimal places
e.g.: 0.0270 (0.0669) -> 0.027 (0.067)
4.9158 (1.3829) -> 4.9 (1.4)
> add a vertical line to separate Log(Inorganic) models from Log(Organic) models
--------
1.5 Table 3
> very poor models... R² < 0.500
--------
1.6 Explain used symbols...
e.g. in Table 4
> L.log (P) ?
> AR(2)?
> Sargan ?
--------
1.7 in line 255
> "House-man test" -> "Houseman test"
##################################################
2. Scientific issues...
2.1 in all presented Tables...
> most model parameters are irrelevant (with no statistical meaning...)
Why to consider that models?
--------
2.2 Some problems in classifying parameter relevance...
e.g. in Table 5
in Model 1 .. Rate = -3.7569(1.7483) -> TV(Rate=0) = 1.82 -> p-value(gl=325) = 0.070 (between 0.10 and 0.05)
t^b_0.10(325) = 1.65 ; t^b_0.05(325) = 1.97 ; t^b_0.01(325) = 2.59
--------
2.3 in Table 5 caption... Rate as the dependent variable?
- Which model is this?
- How can it be?
- Is there a unique value for Rate?
--------
2.4 Tables 3 and 5
> very poor models (R² < 0.500)
- do you still consider those models
##################################################
3. Questions to authors:
3.1 Explain the meaning of "Prob > F" (e.g. in Tables 3 and 5)
--------
3.2. Explain the meaning of "AR (2)" (e.g. in Tables 4 and 6)
--------
3.3 All work was written in terms of "data fitting"...
Best models are able to FITT and PREDICT results...
- Have you preformed any Cross-Validation?
- Why not?
##################################################
Reviewer 3 Report
The manuscript is interesting and sufficiently well organized. The authors carefully treated the subject. Overall impression is that this is a thoroughly studied and written article.
Lines 116-118 I don't think this paragraph is necessary.
Line 221 Point before the start of the new sentence.
Lines 401-406 "There is" should be moved or deleted from the Conclusions in my opinion.
Good luck.
The manuscript is interesting and sufficiently well organized. The authors carefully treated the subject. Overall impression is that this is a thoroughly studied and written article.
Lines 116-118 I don't think this paragraph is necessary.
Line 221 Point before the start of the new sentence.
Lines 401-406 "There is" should be moved or deleted from the Conclusions in my opinion.
Good luck.
Round 2
Reviewer 1 Report
Line 15: should be 'water' pollution instead of 'the water pollution'
Line 25: Before indicating what empirical research suggests should be done, there should be an overall conclusion of the findings themselves. What can we can conclude from these findings in and of themselves before deciding what next steps ought to be.
Line 39: If 'on one hand' is included (which ought to be 'on the one hand') there should be 'on the other hand' later on within a sentence or paragraph. I would suggest removing and rephrasing.
Lines 46-47: This sentence doesn't make sense. It suggests an increase from 24.311 to 735.32 in one year (2015). If this has happened, this is significant and needs to be made clearer. If this is an error, it will need to be corrected.
Line 56: more explanation/setting up of environmental governance needed.
Line 86: should be 'a positive effect'.
Line 90: who is 'they'? Unclear.
Line 125: doesn't sound right to say 'relevant references'.
Line 129: 'some discussion' is vague. Authors need to be more specific here and say, e.g. 'discussion focused on strategies'. Indefinite language ought to be avoided.
Line 134: now suddenly, it says research covers up to 2020 and not up to 2015? It seems like these five years have suddenly been covered in quite a short period of time. Has this had much of an impact on the conclusions?
Line 475: grammatical errors.
Some references are included in the body itself and others are footnoted. There is a need for consistency.
Reviewer 2 Report
Thank you for your effort in correcting your work.
Author Response
Thank you for your comment of "Thank you for your effort in correcting your work."